# Structure-guided discovery of anti-CRISPR and anti-phage defense proteins

Ning Duan[1,2], Emily Hand[1,2], Mannuku Pheko[1], Shikha Sharma[1] & Akintunde Emiola [1] ✉

Bacteria use a variety of defense systems to protect themselves from phage infection. In turn, phages have evolved diverse counter-defense measures to overcome host defenses. Here, we use protein structural similarity and gene co-occurrence analyses to screen >66 million viral protein sequences and >330,000 metagenome-assembled genomes for the identification of anti-phage and counter-defense systems. We predict structures for ~300,000 proteins and perform large-scale, pairwise comparison to known anti-CRISPR (Acr) and anti-phage proteins to identify structural homologs that otherwise may not be uncovered using primary sequence search. This way, we identify a *Bacteroidota* phage Acr protein that inhibits Cas12a, and an *Akkermansia muciniphila* anti-phage defense protein, termed BxaP. Gene *bxaP* is found in loci encoding Bacteriophage Exclusion (BREX) and restriction-modification defense systems, but confers immunity independently. Our work highlights the advantage of combining protein structural features and gene co-localization information in studying host-phage interactions.

Bacteriophages (phages) are considered the most abundant viruses on earth, and they infect nearly half of all sequenced bacterial genomes[1-3]. They are a significant threat to bacteria populations and can cause up to 40% of lysis in habitats such as oceans[4,5]. Bacteria and phages are locked in a perpetual evolutionary warfare that has led to multiple defense and counter-defense measures[6,7].

In bacteria, the most common anti-phage defense mechanisms are adaptive immune systems conferred by RNA-guided CRISPR-Cas[8,9], abortive infection (Abi) and toxin-antitoxin (TA) systems to induce cell death or dormancy upon infection[10,11], and innate systems such as restriction-modification (RM) and cell-surface modification to prevent phage entry[2,12,13]. Recently, new systems with novel mechanisms of action have been discovered[10,14-17]. These include systems that involve production of antiviral compounds[18,19], systems relying on second messenger signaling[20,21], and reverse-transcription of RNA based systems[16,22]. On the other hand, phages have evolved several mechanisms to circumvent host defenses such as anti-RM and anti-CRISPRs (Acr) proteins[7,23]. Acrs are the best studied anti-defense systems and were originally discovered in 2013 from phage genomes infecting

strains of *Pseudomonas aeruginosa*[24]. Since then, over 100 Acrs inhibiting a wide diversity of CRISPR types have been discovered from both phages and mobile genetic elements. Acrs typically inactivate CRISPR systems by directly interacting with Cas proteins to prevent DNA cleavage or target DNA binding[7]. In addition, some Acrs possess enzymatic activities to modify Cas proteins post-translationally[25,26].

There are on-going efforts to identify anti-phage defense and counter-defense systems[10,14-16,27]. An understanding of the "arms race" between phages and its host will not only shed light on their coevolutionary conflict, but also provide translational insights for modern therapeutics. For instance, Acr proteins bring the possibility of precise regulation of the CRISPR system in genome editing[7,28]. Furthermore, phages engineered to contain Acrs were recently shown to suppress infection of antibiotic-resistant *P. aeruginosa*[29]. Lastly, the growing interest in modulating the microbiome using microbe-specific inhibitors makes the prospect of phage therapy exciting[30]. Therefore, identifying the various machineries utilized by bacteria to defend themselves, and the repertoire of phage proteins to overcome these defenses, will be instrumental to these endeavors.

[1]Microbial Therapeutics Unit, National Institute of Dental and Craniofacial Research, National Institutes of Health, Bethesda, MD, USA. [2]These authors contributed equally: Ning Duan, Emily Hand. ✉e-mail: akintunde.emiola@nih.gov

Most recent anti-phage discoveries rely on observations that defense systems co-localize in bacteria genomes, forming so-called "defense islands"[14–16]. Still, types of defense systems and nature of defense island remain poorly understood[15]. In addition, due to the wide diversity of phages and their continuous evolutionary battle with host defenses, there are many defense systems yet to be identified[16]. Similar efforts to uncover new Acrs are challenging due to the poor sequence similarity between known Acrs[31]. Instead, scientists turned to a "guilt-by-association" (GBA) approach based on observations that Acrs are often located next to helix-turn-helix (HTH)-containing anti-CRISPR–associated (*aca*) proteins[27,32]. This approach, however, requires a known *aca* when other Acrs are not already available to begin the search. Recently, multiple machine learning methods[31,33,34] have been developed that have greatly expanded Acr discoveries but are prone to false discoveries relative to GBA approach.

Unlike sequence-based homology calling, structural homology can be retained across long evolutionary timescales[35]. For instance, it was recently shown that annotation of metagenomic proteins can be significantly enhanced by up to 70% by incorporating structural features[35,36]. Therefore, the next frontier of protein homolog detection in the study of bacteria-phage warfare is structure-based, which will enhance the sensitivity for understanding the breadth of these families. To this end, we set out to uncover previously unidentified anti-phage and phage counter-defense systems by analyzing >66 million phage proteins and >330,000 metagenome-assembled genomes (MAGs). Using a combination of structural and genomic co-localization features, we identified and demonstrated the activity of an Acr protein and an anti-phage defense system. Altogether, our work expands the repertoire of defense and counter-defense systems and highlights new strategies for studying the on-going evolutionary warfare between phages and its host.

## Results

### Identification of putative anti-crispr proteins using structural features

To identify Acrs in phage genomes, we began by retrieving ~66.5 million proteins from Integrated Microbial Genomes Virus database (IMG/VR)[37]. We excluded large proteins because over 90% of known Acrs contain less than 200 amino acids[38] (Fig. 1A, Supplementary Data 1). To reduce computational load associated with analysis of such large dataset, we performed protein clustering to remove redundancy resulting in ~15 million protein clusters (Fig. 1A). Because most experimentally validated Acrs are acidic[38], we excluded hits with iso-electric point (pI) values greater than 7. Finally, we only selected clusters with no sequence homology to known Acrs and having no hits in the Conserved Domain Database (CDD)[39] to enable discovery of extremely divergent candidates. The resulting ~7 million protein clusters served as our non-redundant, refined protein catalog for further analyses.

Though, most discovered Acrs are non-homologous, they however, inactivate CRISPR-Cas systems through similar mechanisms which include direct interaction with CRISPR-Cas machinery or modification of Cas protein[7]. Consequently, we hypothesized that while Acrs may share little or no sequence similarity, they probably share conserved structural features. To test this hypothesis, we randomly selected 285,000 clusters from our protein set and predicted their structures using AlphaFold2[40]. Predicting structures for only a subset was necessary because of the computational constraint associated with such a huge dataset. We then performed pairwise structure alignment to find similarities between putative Acr candidates and experimentally-determined Acr structures. We used the TM-score provided by US-align[41] as a measure of structure similarity. The TM score – ranging between 0 and 1 – measures the degree of match of the overall (backbone) shape of two structures[40]. Typically, proteins with TM-score ≥ 0.6 are highly structurally similar[42]. For each reference Acr, we selected hits with the highest TM-score. In total, we identified structural homologs for 8 known Acrs, encompassing 6 anti-CRISPR-types (V-A, I-F, I-E, II-A, I-D, and II-C) (Fig. 1B, Supplementary Fig. 1, Supplementary Data 1 and 2).

### An anti-crispr protein from *Bacteroidota* phage inhibits Cas12a

Among the putative Acrs, one candidate was of interest due to its high structural similarity (TM-score = 0.84) with AcrVA5 and no similarity in amino acid sequence estimated by BLASTP (Fig. 1B, Supplementary Fig. 2A, Supplementary Data 2). AcrVA5 inactivates type V CRISPR-Cas system by acetylation of Cas12a protein[25]. We purified our putative Acr and tested its ability to inhibit Cas12a-mediated DNA cleavage in vitro. We began by incubating our candidate with Cas12a (LbCas12a) at different molar ratios for 10 min, and then added CRISPR RNA (crRNA) and dsDNA, respectively, to the mixture. After an hour of incubation, we observed a near total cleavage inhibition at Acr:LbCas12a ratio of 2.5:1, and a complete inhibition at ratios of 5:1 (Fig. 2A). To assess whether our candidate also inhibits Cas12a via acetylation, we generated point mutations in the small region that aligned, albeit poorly, to AcrVA5 based on BLASTP (Supplementary Fig. 2A). Interestingly, this alignment region is predicted to be important for acetyl-CoA recognition in AcrVA5[25]. Of the three generated mutants, the G75R and T78R mutations completely abolished the anti-crispr ability of our candidate (Fig. 2B).

Our candidate Acr is located in a 109 Kb phage contig recovered from sheep rumen microbial communities. To identify potential host lineage, we used BLAST hits of protein-coding genes in the contig to infer phylogenetic distribution. Though most genes (84%) could not be assigned to a taxon, ~10% had hits to *Bacteroidota* phylum (Supplementary Fig. 2B). Using a similar approach, we identified homologs in predominantly *Bacteroidota* phages (~74%), and a small number in *Proteobacteria* (2%) and Actinobacteria (3%) phages (Fig. 2C). Thus, we named this protein AcrVA5$_{Bsp}$ in line with recommended naming

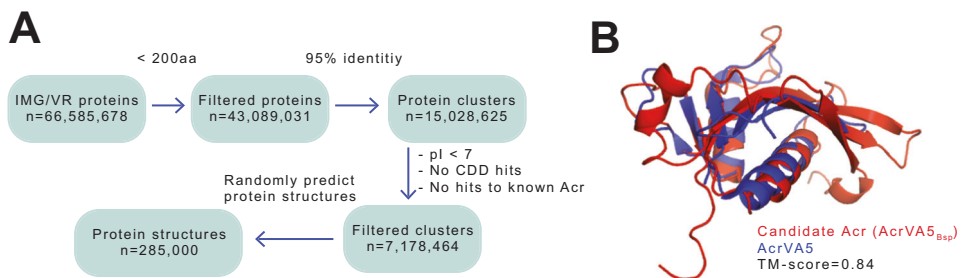

**Fig. 1 | Identification of putative Acrs using structural features. A** Schematic depiction of pipeline used to identify Acr proteins. **B** AlphaFold2 model of newly identified Acr (*AcrVA5$_{Bsp}$*, in red) superposed to the experimental structure of AcrVA5 (in blue). The TM score (0-1) is used to estimate the topological similarity of protein structures. TM scores > 0.6 are structurally similar.

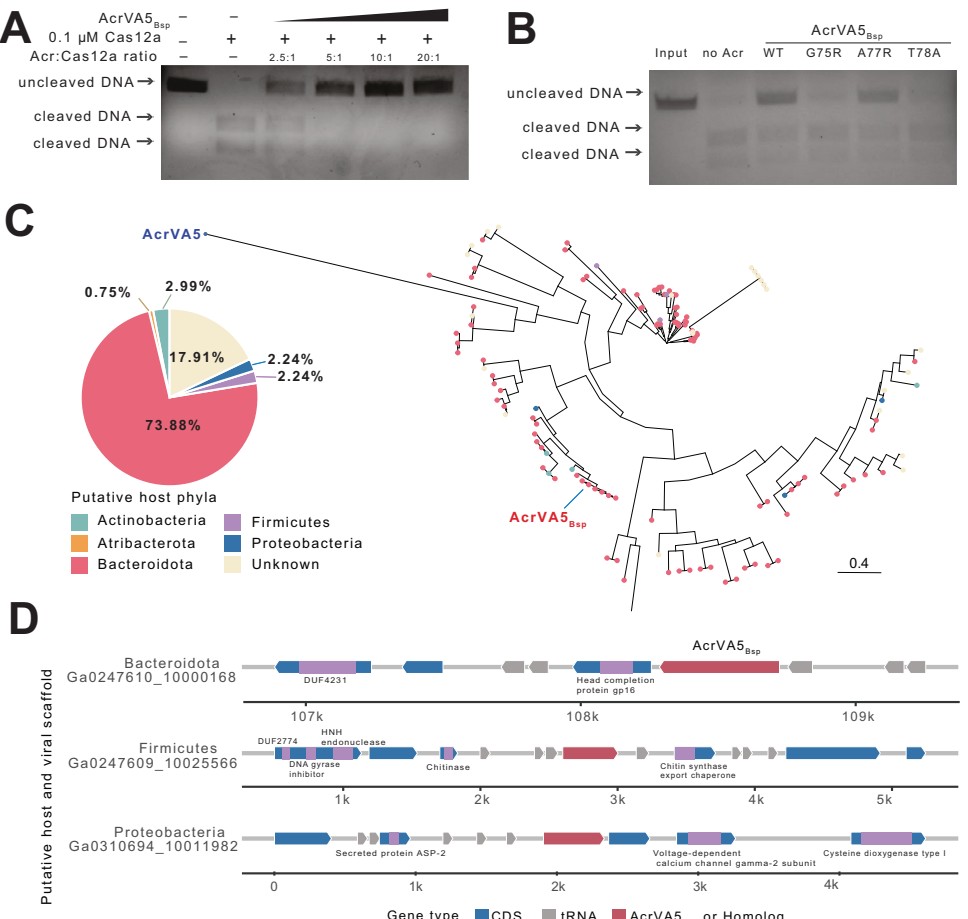

**Fig. 2 | Newly identified Acr protein inhibits Cas12a. A** Agarose gel showing LbCas12a-mediated dsDNA cleavage in the presence of increasing amounts of AcrVA5$_{Bsp}$. Acr concentrations used are 0.25, 0.5, 1, and 2 μM. The image is representative of triplicate. **B** Agarose gel showing dsDNA cleavage inhibition using wild-type (WT) and mutant AcrVA5$_{Bsp}$. Acr concentration used is 0.4 μM. Acetyl-CoA was included in the assays. The image is representative of duplicate. **C** Phylogenetic distribution of AcrVA5$_{Bsp}$ across IMG/VR sequences. The pie chart shows the percentage distribution of AcrVA5$_{Bsp}$ homologs across predicted host phyla. Likewise, branch tips on the right are colored by predicted host of origin. For reference purposes, AcrVA5 was included during tree generation. **D** Examples of genomic loci of AcrVA5$_{Bsp}$ (top) and homologs in IMG/VR scaffolds. The indicated taxa are the putative host phyla. Domains of neighboring genes are colored purple. Source data are provided as a Source Data file.

convention[43]. In addition, current Acrs targeting type V CRISPR systems were discovered from *Moraxella* phages (Supplementary Data 1). To our knowledge, this is the first report of Cas12a inhibitors from *Bacteroidota* phages. We also purified a sequence homolog of AcrVA5$_{Bsp}$ derived from a Firmicutes phage and found it effectively inhibits LbCas12a; although it is less potent than AcrVA5$_{Bsp}$ (Supplementary Fig. 2C).

We next examined the ability of common Acr-discovery tools to accurately assign AcrVA5$_{Bsp}$. First, AcrVA5$_{Bsp}$ does not co-occur with an HTH-containing *aca* protein and will, therefore, not be detected using "guilt-by-association" approach (Fig. 2D). In addition, we found no hits in Anti-CRISPRdb, a database of experimentally validated and predicted Acr candidates[44]. Other programs such as AcrNET[34], AcrRanker[45], AcrHub (HMM-based predictor)[46] failed to detect AcrVA5$_{Bsp}$. PaCRISPR[33], however, correctly assigned AcrVA5$_{Bsp}$ albeit with low confidence (53% probability) (Supplementary Data 3). Together, these findings highlight the advantage of incorporating structural features in identifying the breadth of Acr families.

### Some putative Acrs share structural similarities with multiple known Acrs

Previously published Acrs were mainly discovered using primary sequence searches and we reckoned that some of these Acrs classified as unique may not be distinct, but instead represent remote homologs.

We performed pairwise structural comparison between all known Acrs and observed that while the majority are distinct, a few are indeed remote homologs (Fig. 3A). For example, AcrIB and AcrVA5 are structurally similar, despite having no sequence similarity (Fig. 3A). Another key advantage of structure-based approach is that, not only will it broaden the sequence space of a given Acr family that may inhibit different orthologs of the same Cas family, it is also likely to identify inhibitors of different Cas families that can be repressed using similar mechanisms without the limitation of sequence constraint. For instance, we found AcrIE8 and AcrIC7 share strong structural similarities which may explain why the latter effectively inhibits I-C and I-E Cas families (Fig. 3B)[47]. In addition, our structure-based approach was able to accurately identify AcrIE4-F7 as a fusion of two Acr proteins (AcrIE4 and AcrIF7) (Fig. 3A).

We next sought to identify putative Acrs from our catalog that may either be a fusion of two known Acrs or share structural similarities with multiple Acrs. Using a TM score cutoff of 0.6, we identified 920 putative Acr candidates with strong structural similarities to 2 different Acrs (Supplementary Data 4). For example, one candidate appears to be a fusion of structural homologs of AcrIC5 and AcrIIC4 (Fig. 3C, D).

Finally, to expand the scope of phage counter-defense systems from our predicted structures, we searched for structural homologs to other known anti-defense systems. We downloaded sequences for

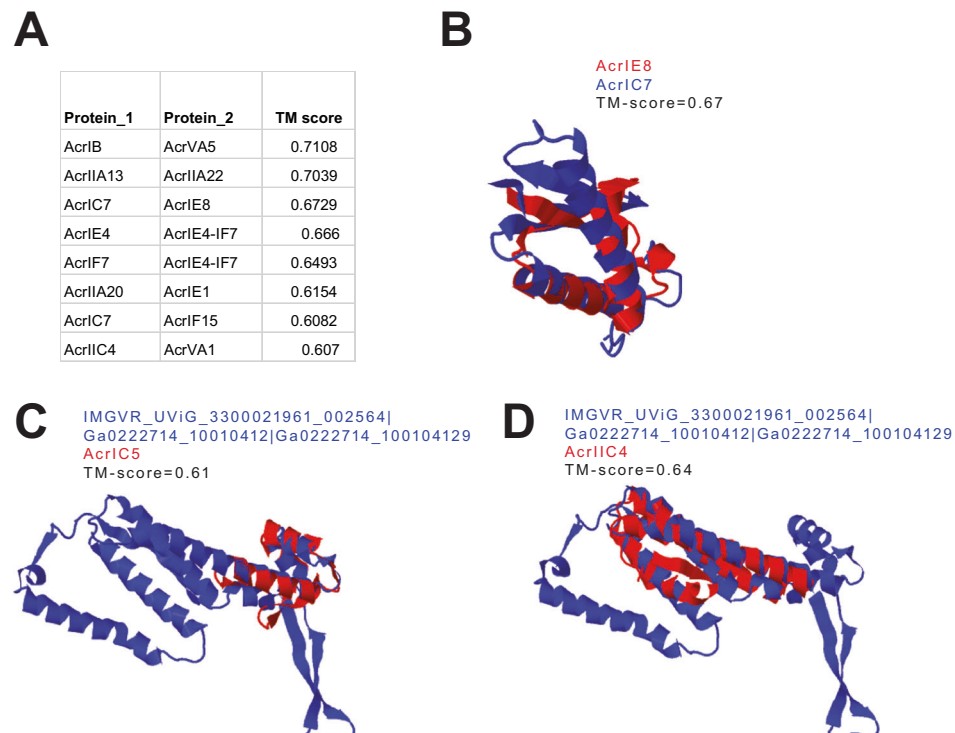

**A**

| Protein_1 | Protein_2 | TM score |
|-----------|-----------|----------|
| AcrIB | AcrVA5 | 0.7108 |
| AcrIIA13 | AcrIIA22 | 0.7039 |
| AcrIC7 | AcrIE8 | 0.6729 |
| AcrIE4 | AcrIE4-IF7 | 0.666 |
| AcrIF7 | AcrIE4-IF7 | 0.6493 |
| AcrIIA20 | AcrIE1 | 0.6154 |
| AcrIC7 | AcrIF15 | 0.6082 |
| AcrIIC4 | AcrVA1 | 0.607 |

**B**

AcrIE8
AcrIC7
TM-score=0.67

**C**

IMGVR_UViG_3300021961_002564|
Ga0222714_10010412|Ga0222714_100104129
AcrIC5
TM-score=0.61

**D**

IMGVR_UViG_3300021961_002564|
Ga0222714_10010412|Ga0222714_100104129
AcrIIC4
TM-score=0.64

**Fig. 3 | Known and putative Acrs sharing structural similarities with multiple Acrs. A** Pairwise structural comparison between all known Acr proteins. Acr pairs with TM score > 0.6 are considered to be structurally similar. **B** An example of AlphaFold2 model of AcrIE8 (in red) superposed to AcrIC7 (blue). **C, D** An example of a putative Acr protein (in blue) sharing structural similarities to both (**C**) AcrIC5, and (**D**) AcrIIC4.

anti-CBASS, anti-RecBCD, anti-restriction, and anti-Thoeris proteins from NCBI database, predicted their structures using AlphaFold2[40], and performed pairwise structural alignment to our cataloged protein structures. Similarly, using a TM score cutoff of 0.6, we identified hits to anti-restriction, anti-CBASS, anti-RecBD proteins despite sharing no sequence similarity (Supplementary Fig. 3). Altogether, these findings highlight the advantage of incorporating structural features in identifying extremely diverse protein homologs of phages involved in evading bacteria defenses.

### Domain-independent identification of single-gene anti phage defense systems

Having identified a phage counter-defense Acr protein, we then set out to uncover new strategies through which bacteria provide defense against phage infection. Since many anti-phage genes cluster in defense islands, multiple studies have uncovered new defense systems by searching for protein domains enriched in these islands[14–16]. In other words, a domain that consistently occurs with several known defense genes is likely to be involved in defense. We took a similar approach with modifications to prioritize rare single-gene defense systems and proteins with either an unknown domain or domains not previously implicated in defense.

We began by retrieving ~330,000 MAGs from the Global Microbial Genome Bins (GMBC)[48] and the Genomic catalog of Earth's Microbiomes (GEM)[49]. These MAGs were assembled from diverse human, animal, and environmental microbiomes. Next, we identified known anti-phage genes in MAGs using PADLOC, a tool that annotates defense systems based on sequence homology and genetic architecture[50]. For two defense systems separated by 1 – 15 Kb, we retrieved all proteins found between both systems resulting in 390,149 proteins from 83,926 MAGs. These were subsequently clustered at 50% identity to generate 66,285 non-redundant clusters (Fig. 4A). Of these, only ~57% had annotated protein domains based on hits to CDD.

We hypothesized that a large subset of clusters with unknown domains will be associated with defense (given their proximity to known defense genes) and likely share similar structural features with other characterized single-gene defense systems. To test this hypothesis, we selected candidates with no CDD hits and predicted the structures of 10,620 proteins using AlphaFold2[40]. We then performed pairwise structure alignment to find similarities with single-gene defense systems previously reported in Vassallo et al.[10]. Five candidates with TM scores > 0.65 and having no significant amino acid similarities with reference sequences, were selected for experimental validation (Fig. 4B, Supplementary Fig. 4A, Supplementary Data 5).

### A rare gene associated with BREX defense system confers phage resistance independently

We cloned our top candidates into a plasmid vector, transformed *E. coli* cells, and challenged transformants with T4 and T7 lytic phages. We were unable to validate two candidates (ORF5348 and ORF8840) because gene expression induced cellular toxicity in the absence of phage infection. Two additional candidates (ORF8850 and ORF1717) provided no defense against both phages (Supplementary Data 5). However, expression of ORF376 (subsequently renamed BxaP as described below), a gene identified in an *Akkermansia muciniphila* MAG, provided a thousand-fold protection against T4 phage and moderate protection against T7 phage (i.e. smaller plaque sizes) (Fig. 4C, Supplementary Fig. 4B). A mutation (P121A) in the N-terminus region of ORF376 significantly reduced protection against T4 phage (Fig. 4C, Supplementary Fig. 5A). In our catalog, ORF376 belongs to a cluster with identical proteins (i.e. cluster members having 100% sequence identity) and co-occurs with type 3 Bacteriophage Exclusion (BREX) anti-phage system[50] (Fig. 5A). We found this interesting because PD-T7-3, which is structurally similar to ORF376 (Fig. 4B), does not appear in defense islands[10]. The canonical type 3 BREX locus comprises 6 genes: *BrxF, BrxC/PglY, PglXI, BrxHII, PglZ*, and *BrxA*[51]. Surprisingly, ORF376, not part of the typical system, is inserted

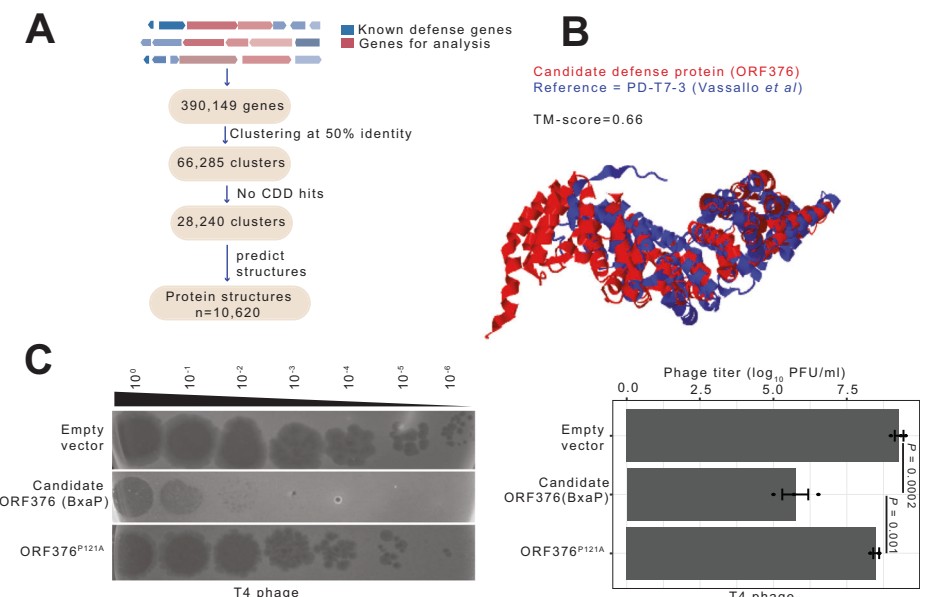

**Fig. 4 | A defense protein, BxaP, confers protection against phages. A** Schematic overview of pipeline used to identify defense systems. Genes found within known defense systems are retrieved and subjected to multiple filtration steps. The domain-independent approach involves predicting structures of >10,000 uncharacterized proteins lacking functional annotation (i.e., no CDD hit), and performing structural alignment to known single-protein defense systems to identify distant homologs. **B** AlphaFold2 model of a candidate defense protein (ORF376, in red) superposed to the predicted structure of a known single-protein defense system (PD-T7-3, in blue). **C** A 10-fold serial dilution plaque assay of T4 phage on *E. coli* strain containing either an empty vector, or overexpressing WT or mutant BxaP. The image is representative of triplicate. The barplot on the right shows the efficiency of plaquing of T4 phage infecting *E. coli* harboring an empty vector, or overexpressing WT or mutant BxaP. Experiments were conducted in triplicates and error bars represent SEM. Test of normality was conducted using Shapiro-Wilk test and statistical significance was determined using ANOVA. The P-value is two-sided. Source data are provided as a Source Data file.

between *PglXI* and *BrxHII* (Fig. 5A). Among 7308 MAGs with a BREX system, only 21 contained ORF376, indicating a prevalence rate of 0.29% (Fig. 5B). Similarly, we searched for homologs in non-redundant (nr) NCBI protein database and identified a few hits in distantly related species (Supplementary Fig. 5B, Supplementary Fig. 6). We also searched for homologs in phage genomes because several defense genes were recently uncovered in prophages[10]. Using BLASTP search against IMG/VR database[37], we identified hits in 5 genomes. Surprisingly, homologs were found incorporated only in type-I RM systems in these phages (Supplementary Fig. 5C). Furthermore, though our test *E. coli* strain lacks BREX, it possesses an active type-I RM system. We therefore examined if ORF376 activity is retained in cells lacking both systems. We observed ORF376-mediated defense in an *E. coli* strain (DH10B) lacking both BREX and type-I RM systems (Supplementary Fig. 5D).

To determine a likely mode of action, we used HHpred[52] to detect remote structural similarities to known protein domains/families. The N-terminus of ORF376 had strong hits to DUF4145 domain (98.5 probability, evalue 2.6e-6) which is found in some RM systems whereas, the C-terminus had a weak structural hit to eEF3-like HEAT domain of *Saccharomyces cerevisiae* Gcn1 protein (94.6 probability, evalue 14) (Fig. 5B). The eEF3-like HEAT domain, and Gcn1 in general, interacts with ribosomes and participates in the repression of global protein synthesis[53]. Phage defense via global gene repression is characteristic of Abi defense systems[13]. Typically, in experimental assays involving Abi systems, phage resistance is usually observed at low multiplicity of infection (MOI) in which bacteria concentration exceeds phages whereas, high MOI results in massive cell death[10]. To test whether ORF376 works through a similar mechanism, we infected *E. coli* cells with varying MOIs of T4 phage and monitored growth dynamics over time. At all MOIs tested, we observed complete growth inhibition in cells lacking a defense system or harboring a mutant ORF376. However, at MOI of 0.001, cultures with ORF376-defense system grew similar to control whereas, no growth was observed at an

MOI of 2 (Fig. 5C). We measured phage titers 3 hours post-infection (MOI of 2) and found phage replication did not occur in ORF376 cells (Fig. 5D). Therefore, the inability of ORF376 cells to grow was not due to viable virions, suggesting a likely Abi defense strategy. Consequently, we renamed ORF376 as "**B**REX/RM-associated **A**nti-phage **P**rotein" (BxaP). Together, our results suggest BxaP-mediated defense is generally associated with defense systems that confer direct immunity but can also provide protection independently.

## Discussion

The streamlined methodological approach typically taken to study host-phage evolutionary warfare has made it difficult to identify rare host anti-phage and phage counter-defense systems. In this work, we took an alternate approach by combining protein structural similarity screening and gene co-occurrence information to uncover an Acr protein and a protein involved in phage defense.

Though it is believed there are far more Acrs than CRISPR-Cas systems in nature, the identification and characterization of CRISPR-Cas systems greatly outnumber Acr discovery[54] and this is partly due to poor sequence conservation among Acrs. Because our approach is independent on primary sequence alignment, it is agnostic to genomic features such as presence of a flanking HTH-containing *aca* gene or the presence of CRISPR-Cas system containing self-targeting spacers which many Acr discovery method rely on. Incorporating structural features in our approach enabled detection of remote Acr homologs and validation of a candidate, AcrVA5$_{Bsp}$.

Though AcrVA5$_{Bsp}$ shares no sequence similarity with known Acrs, the high structural similarity with an acetyltransferase, indicates an inhibitory mechanism involving post-translational modification of Cas12a. Indeed, mutation in residues that abolished acetyl-CoA recognition in AcrVA5 also abolished AcrVA5$_{Bsp}$ anti-crispr activity. Surprisingly, AcrVA5$_{Bsp}$ homologs in phages infecting hosts from different phyla are often located next to multiple tRNAs (Fig. 2D). It is believed that phage-encoded tRNAs play a role in counter-defense,

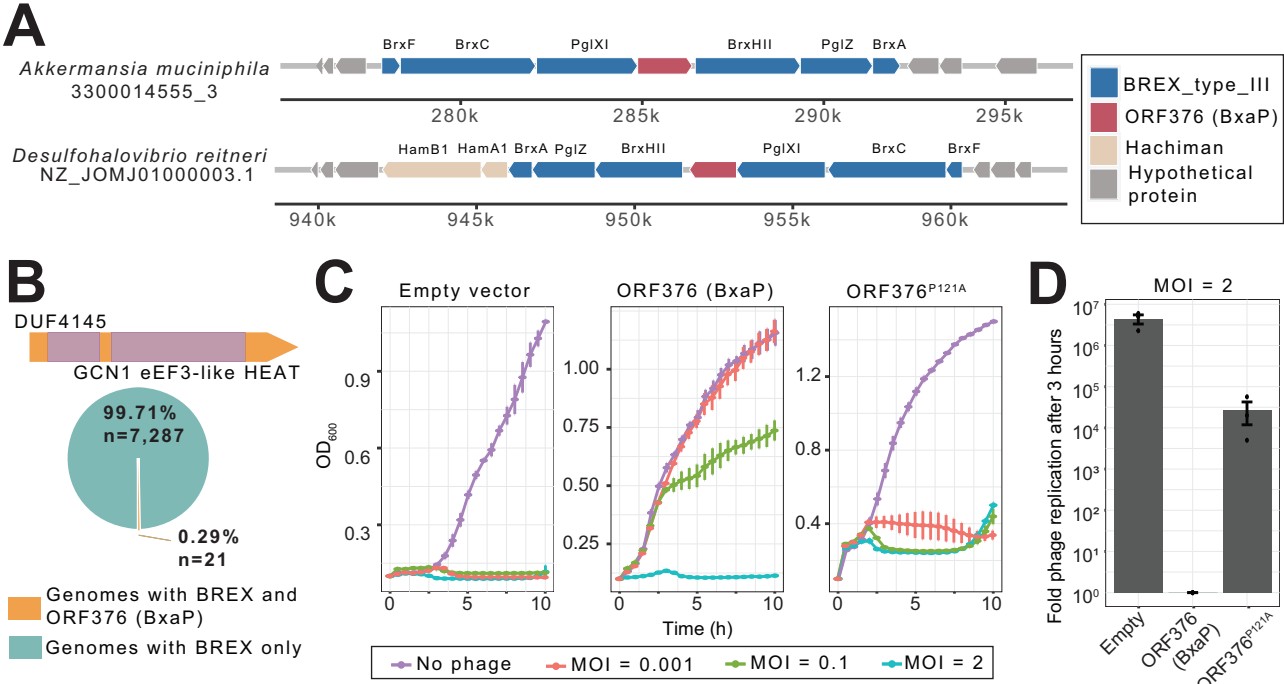

**Fig. 5 | BxaP confers immunity likely through abortive infection. A** BxaP is incorporated in type 3 BREX defense systems. Examples of genomic loci containing BxaP (top) and homolog in bacterial genomes. Genome/MAG identifier is shown below each taxon. **B** Distribution of BxaP-containing BREX systems in MAGs. The top image shows HHpred predicted domains in BxaP. **C** Growth curves of BxaP-containing *E. coli* in comparison with control (empty vector) and mutant-BxaP under varying T4 phage concentrations. Experiments were conducted in triplicates, and error bars represent SEM. **D** T4 phage replication dynamics in empty vector, BxaP-mutant, and BxaP-containing cells. Experiments were conducted in triplicates and error bars represent SEM. Source data are provided as a Source Data file.

allowing evasion of bacteria defenses that rely on depleting host tRNAs to prevent phage infection[55]. Since Acrs sometimes co-localize with other antagonists of host defense[27], AcrVA5$_{Bsp}$ may be situated in an anti-defense hotspot. Indeed, other genes flanking AcrVA5$_{Bsp}$ contain domains, such as DUF4231 and HNH endonuclease, that have been reported to be enriched in anti-phage defense islands[15] (Fig. 2D). These suggest that this genomic region contains genes that provide protection to prophages and protect bacterial host from superinfection.

The newly identified anti-phage systems, BxaP, is a rare, less prevalent, defense protein that is incorporated into type-3 BREX and RM systems but confers immunity independently. We identified homologs in few and uncommon genomes such as *Desulfohalovibrio reitneri* isolated from suboxic zone of a hypersaline lake[56], *Tepidanaerobacter syntrophicus* isolated from thermophilic digested sludges[57], and *Inmirania thermothiophila* isolated from a shallow-sea hydrothermal vent[58] (Supplementary Fig. 5B). Thus, BxaP may have been recently acquired by bacteria in hostile environments to overcome the consequences of phage infection. Alternatively, evolution of viral counter-defense may have eliminated the BxaP system from many sequenced genomes.

How bacteria cells recognize infection and trigger BxaP remains to be established. Since DUF4145 (present in BxaP) is predicted to be a member of the HEPN superfamily domain[59], it is also unclear whether BxaP transitions between an inactive and active state via a conformational change, similarly to other HEPN-containing toxins[59]. We also do not know if/how phages may escape BxaP defenses. We tried to evolve T4 phage but were unable to identify escape mutants even after multiple rounds of phage passage.

A dual direct/Abi defense strategy is not uncommon and has previously been observed with Cas12a2 which elicits suicide by activating indiscriminate nucleases[60], activation of PrrC toxin upon inhibition of RM systems[61], and PARIS mediated Abi defense upon BREX/RM inactivation[17]. In addition, the phage defense system PD-λ−5, is thought to be a highly compacted prophage version of a dual RM/Abi

system[10]. The PARIS system, however, is conceptually similar to BxaP. Nevertheless, both systems have different protein domains and likely utilize separate mechanisms to provide defense. Given the similarities between BREX and RM systems, and that both systems confer phage resistance via direct immunity, an abortive-like mechanism potentially acts as a complementary strategy to protect uninfected populations when other defense machineries have failed.

In conclusion, our work extends the range of phage and host defense arsenal and highlights new strategies to identify defense and counter-defense systems. A major limitation of our structure-based approach is the requirement of a characterized defense protein to act as reference, and the inability to identify multi-gene defense systems. Nevertheless, we demonstrate our method has the advantage of identifying rare and less prevalent systems which may be elusive to conventional computational approaches. Future discoveries will not only expand our basic understanding of predator-prey arms race, but also lay a foundation to develop more molecular tools for biotechnological and biomedical uses such as precise regulation of the CRISPR system in genome editing and targeted microbial phage therapy.

## Methods

### Bioinformatics pipeline for Acr discovery
We downloaded 66,585,678 viral proteins from IMG/VR V3 (version 2020-10-12_5-1)[37] and excluded proteins > 200 amino acids. The resulting 43,089,031 small proteins were clustered at 95% identity using USEARCH (v. 11.0.667)[62] to generate 15,028,625 clusters, with centroids being the representative sequence. To facilitate discovery of remote Acr homologs, we performed BLASTP (BLAST v. 12.13.0) search to 100 experimentally-validated Acrs and excluded hits having > 30% identity or evalue < 0.1. Only 14,098,963 clusters passed this threshold. Because most validated Acrs are acidic, we determined the isoelectric point (pI) for representative sequences using IPC 2.0. software[63] and discarded hits with pI values > 7. Next, the streamlined sequences

($n$ = 8,662,777) were searched for protein families against the Conserved Domain Database (CDD)[39] (evalue cutoff = 0.01), and we subsequently excluded hits to prioritize candidates with unknown functional properties. The final set of 7,178,464 sequences served as our non-redundant, refined protein catalog. We randomly selected 285,000 proteins for protein structure prediction and performed pairwise structure alignment to find similarities between our protein catalog and previously published Acrs as described below.

## Computational prediction of single-gene defense systems

We retrieved 331,144 MAGs from the Global Microbial Genome Bins (GMBC)[48] ($n$ = 278,629) and the Genomic catalog of Earth's Microbiomes (GEM)[49] ($n$ = 52,515) and predicted known defense systems using PADLOC (v. 1.1.0)[50]. For two defense systems separated by 1–15 Kb, we retrieved all proteins located between both systems, resulting in 390,149 proteins from 83,926 MAGs. We then clustered these proteins at 50% identity using USEARCH[62] to generate 66,285 clusters, with centroids as representative sequences.

For our domain-independent approach, we retrieved sequences with no hits in CDD ($n$ = 28,240) and discarded candidates <200 amino acids. This is because most single-gene defense systems tend to be large proteins[10]. We predicted the protein structures of the resulting 10,620 proteins and performed pairwise structure alignment to find similarities with single-gene defense systems previously reported in Vassallo et al.[10]. A detailed description on protein prediction and pairwise structure alignment is described below.

## Prediction of protein structures and pairwise comparison

The structures of phage and bacterial proteins were predicted using AlphaFold2 (v. 2.1.2) (−model_preset = monomer)[40]. For each protein, 5 models were generated, and we selected the best model (ranked_0.pdb) determined by the average pLDDT score. We used US-align (v. 20220511)[41] for pairwise structural alignment (using default parameters) and to generate superposed structures which was subsequently visualized using PyMOL (v. 2.6.0)[64].

## Protein annotation and phylogenetic analyses

Unless stated otherwise, protein homologs were determined by BLASTP search using a 30% identity threshold and query coverage > 50%. For remote homology search and domain annotation, proteins were scanned using the online version of HHpred against the PDB, UniProt, and Pfam databases[52].

Phylogenetic analyses were performed by aligning protein sequences using MUSCLE (v. 5.1)[65], and then using maximum likelihood (ML)-based FastTree (v. 2.1.11) with 1000 bootstraps to generate phylogenetic tree[66].

## Identification of remote homologs of other phage counter-defense proteins

Sequences for 46 anti-CBASS, 1905 anti-RecBCD, 911 anti-restriction, and 10 anti-Thoeris proteins were obtained from NCBI database, and their structures were predicted using AlphaFold2[40]. The predicted structures were aligned to the structures of our 285,000 protein catalog using US-align[41]. Alignment pairs with TM-score > 0.6 of shorter sequence and length of shorter sequence >= 70% length of longer sequence were retained for protein sequence alignment with BLASTP. Primary sequence alignment hits with > 30% identity or evalue < 0.1 was removed to ensure detection of remote homologs. In total, we identified 42 phage proteins that have high structural similarity, but no significant sequence similarity, to 41 counter-defense proteins.

## AcrVA5$_{Bsp}$ and AcrVA5$_{Firm}$ purification and in vitro DNA cleavage assay

AcrVA5$_{Bsp}$ and AcrVA5$_{Firm}$ (IMG protein id Ga0247609_100255669) expression and purification was performed by Biomatik. Briefly, *E. coli*

bearing a plasmid containing the gene of interest was induced with 0.2 mM IPTG at 37 °C. After overnight incubation, cell pellets were resuspended in buffer (50 mM Tris, pH 8.0, 300 mM NaCl, 20 mM Imidazole containing 1% Triton X-100,1 mM DTT,1 mM PMSF), followed by sonication. At the same time, Ni-IDA affinity chromatography column was balanced with 50 mM Tris (pH 8.0), 300 mM NaCl, 20 mM Imidazole buffer. Next, the target protein was eluted with different concentrations of imidazole balanced buffer, and each elution component was collected for SDS-PAGE analysis.

CRISPR-RNA and dsDNA were custom synthesized by GenScript. For cleavage assay, LbCas12a (NEB M0653S), acetyl CoA and purified AcrVA5$_{Bsp}$ (or AcrVA5$_{Firm}$) protein were mixed in cleavage buffer (1X r.2.1 NEB buffer) to a total volume of 20 µl and incubated at room temperature for 10 min. 0.5 µl of crRNA was added and incubated for an additional 5 min at room temperature. Lastly, 10 µl of dsDNA (in cleavage buffer) was added and incubated at 37 °C for 1 hr. Thus, each reaction contained 10 nM dsDNA, 100 nM LbCas12a, 200 nM acetyl CoA, 200 nM crRNA, and Acr concentrations of either 0.25 µM, 0.5 µM, 1 µM, or 2 µM in a total volume of 30.5 µl. The reaction was stopped by incubating the mixture at 85 °C for 5 min and the cleavage products were visualized on a 1% agarose gel. See Supplementary Data 6 for sequence information.

## Bacteria culture conditions

Unless stated otherwise, *E. coli* MG1655 and *E. coli* DH10B were routinely grown at 37 °C in LB media. When required, antibiotics were used at the following concentrations: 30 µg/ml gentamicin (or 15 µg/ml in broth). Protein expression was induced with 0.01 % arabinose.

## Plasmid preparation and bacteria transformation

Bacteria and plasmids used in our study are presented in Supplementary Data 7. All putative defense genes were codon optimized for *E. coli* using IDT Codon Optimization Tool, synthesized and cloned into pHERD30T vector by GenScript. Plasmids were transformed into chemically-competent *E. coli* via heat-shock method. To make cells competent, overnight cultures were inoculated 1:100 into 25 ml LB broth, incubated at 37 °C to an OD of ~ 0.4, and pelleted by centrifugation. The pellets were gently resuspended in 20 ml of sterile, ice-cold CaCl$_2$ (100 mM) and allowed to sit on ice for 15 min. Next, washed cells were collected via centrifugation and further resuspended in ice-cold CaCl$_2$ supplemented with 15% glycerol. Aliquots of 50 µl were stored at −80 °C.

To transform cells, 100 ng of plasmid was added to 50 µl competent cells and allowed to sit on ice for 30 min. The mixture was transferred to a 42 °C water bath for 45 s, immediately placed on ice for 2 min, and 950 µl of SOC media was added. After 1 hr of incubation with shaking at 37 °C, 90 µl was spread onto LB agar plates containing appropriate antibiotics and incubated overnight at 37 °C.

## Plaque assay

Plaque assay was conducted as previously described with some modifications[21,27]. Briefly, 200 µl of overnight *E. coli* (containing putative defense system) or control cultures was combined with 4 ml of 0.5% molten top LB agar and the resulting mixture was poured on a pre-warmed LB agar plate containing appropriate antibiotics and inducer. Upon solidification, 5 µl of a 10-fold serial dilution of T4 and T7 phage stock were spotted on the plate. Spots were allowed to dry and incubated upright at 37 °C overnight. Plaque forming units (PFUs) were enumerated by counting the derived plaques.

## Abortive infection assay

Abortive infection assay was performed as previously described with minor modifications[10]. For growth assays, overnight *E. coli* cultures were normalized to an OD$_{600}$ of 1 and diluted 1:100 in fresh LB media supplemented with appropriate antibiotics and inducer. 150 µl was

dispensed in a 96-well plate and 5 μl of phage dilution was added. Growth dynamics ($OD_{600}$) was monitored at 37 °C using Agilent BioTek microplate reader.

To measure phage replication at high MOIs, *E. coli* cultures were grown in LB broth (supplemented with 1 mM $CaCl_2$, antibiotics, and inducer) to an $OD_{600}$ of ~0.2. For assays involving gentamicin, cells were washed to remove antibiotics since aminoglycosides have been reported to reduce phage infectivity[19]. Cells were resuspended in pre-warmed LB (supplemented with 1 mM $CaCl_2$), and infected with T4 phage at an MOI of 2. The control experiment was performed similarly but contained LB broth with no bacteria. After 3 hours of incubation at 37 °C without shaking, PFUs were enumerated by means of the plaque assay and using cells harboring empty vector as indicator strain.

One-step growth assays were conducted similarly with slight modifications. Washed cells were resuspended in LB broth as above and infected at an MOI of 0.05. Samples were collected at regular intervals, serially diluted, and immediately plated on agar plates with empty vector as indicator strain to determine PFUs.

### Reporting summary

Further information on research design is available in the Nature Portfolio Reporting Summary linked to this article.

## Data availability

Phage protein sequences were retrieved from IMG/VR (https://img.jgi.doe.gov/cgi-bin/vr/main.cgi) while MAGs were retrieved from GEM (https://genome.jgi.doe.gov/portal/GEMs/GEMs.home.html) and GMBC (https://gmgc.embl.de/download.cgi). The ~300,000 predicted protein structures in this study can be downloaded from figshare (https://doi.org/10.6084/m9.figshare.24948714.v1). All other necessary data are included in the Supplementary Information. Source data are provided with this paper.

## Code availability

We developed a tool to retrieve uncharacterized genes between known anti-phage defense genes. The software is freely available at https://github.com/EmiolaLab/ExtractGenes[67].

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

## Acknowledgements

We thank Alan Davidson for providing the pHERD30T plasmid. We also thank Brendan Woodworth for providing technical assistance, and Dr. Nadine Samara for fruitful discussions. This work was supported by the NIH NIDCR Intramural Research Program.

## Author contributions

A.E. conceived the study. N.D. and A.E. performed bioinformatics analyses. E.H., M.P. and S.S. performed Acr and phage experiments. N.D., E.H. and A.E. analyzed the data. N.D. and A.E. wrote the manuscript. A.E. supervised the study.

## Competing interests

The authors declare no competing interests.
