## [Peer Review File · Nature Communications]

Structure-guided discovery of anti-CRISPR and anti-phage defense proteinsEditorial Note: This manuscript has been previously reviewed at another journal that is not operating a transparent peer review scheme. This document only contains reviewer comments and rebuttal letters for versions considered at *Nature Communications*.

Reviewer #3 (Remarks to the Author):

Congrats to the authors on a very nice study.

Not sure how the system that is abortive has weakened growth at MOI 0.1, perhaps it is being overwhelmed! Therefore, I would expect at higher MOI it would also look more overwhelmed and perhaps mislead into abortive conclusion. So I would push back on this conclusion but not uncommon to call this abortive in the field even if not fully fleshed out.

Either way, I think paper is a great resource and love the structure guided approaches. Ready to publish I think.

Congrats to the authors on a very nice study.

Not sure how the system that is abortive has weakened growth at MOI 0.1, perhaps it is being overwhelmed! Therefore, I would expect at higher MOI it would also look more overwhelmed and perhaps mislead into abortive conclusion. So I would push back on this conclusion but not uncommon to call this abortive in the field even if not fully fleshed out.

Either way, I think paper is a great resource and love the structure guided approaches. Ready to publish I think.

Thank you!

BxaP is a structural homolog of a reported abortive infection protein. Nevertheless, we have avoided definitive claims of BxaP being an abortive protein but rather, use cautionary wording. In this regard, we renamed BxaP as “BREX/RM-associated Anti-phage Protein” instead of “BREX/RM-associated Abortive Protein”.